# Effects of Resistance Training on Oxidative Stress Markers and Muscle Damage in Spinal Cord Injured Rats

**DOI:** 10.3390/biology11010032

**Published:** 2021-12-27

**Authors:** Natalie de Almeida Barros, Felipe J. Aidar, Anderson Carlos Marçal, Jymmys L. Santos, Raphael Fabricio de Souza, Jainara Lima Menezes, Margarete Zanardo Gomes, Dihogo Gama de Matos, Eduardo Borba Neves, André Luiz Gomes Carneiro, Paulo Francisco de Almeida-Neto, Breno Guilherme de Araújo Tinoco Cabral, Reinaldo Viana Belo Neto, Beat Knechtle, Filipe Manuel Clemente, Enilton Aparecido Camargo

**Affiliations:** 1Graduate Program in Health Science, Federal University of Sergipe (UFS), São Cristovão 49100-000, Brazil; barrosnatalie@yahoo.com.br (N.d.A.B.); enilton.camargo@gmail.com (E.A.C.); 2Graduate Program of Physical Education, Federal University of Sergipe (UFS), São Cristovão 49100-000, Brazil; acmarcal@yahoo.com.br (A.C.M.); jymmys.lopes@gmail.com (J.L.S.); raphaelctba20@hotmail.com (R.F.d.S.); 3Group of Studies and Research of Performance, Sport, Health and Paralympic Sports (GEPEPS), Federal University of Sergipe (UFS), São Cristovão 49100-000, Brazil; 4Department of Physical Education, Federal University of Sergipe (UFS), São Cristovão 49100-000, Brazil; 5Graduate Program of Physiological Science, Federal University of Sergipe (UFS), São Cristovão 49100-000, Brazil; jhaynnara@hotmail.com; 6Laboratory of Morphology and Experimental Pathology, University Tiradentes (UNIT), Aracaju 49010-390, Brazil; guetezanardo@yahoo.com.br (M.Z.G.); reinaldo.viana@souunit.com.br (R.V.B.N.); 7Cardiovascular & Physiology of Exercise Laboratory, University of Manitoba, Winnipeg, MB R3T 2N2, Canada; dihogogmc@hotmail.com; 8Graduate Program on Biomedical Engineering, Federal Technological University of Paraná (PPGEB/UTFPR), Curitiba 80230-901, Brazil; eduardoneves@utfpr.edu.br; 9Department of Physical Education, State University of Montes Claros, Montes Claros 39401-089, Brazil; algcarneiro@gmail.com; 10Department of Physical Education, Federal University of Rio Grande do Norte, Natal 59064-741, Brazil; paulo220911@hotmail.com (P.F.d.A.-N.); brenotcabral@gmail.com (B.G.d.A.T.C.); 11Institute of Primary Care, University of Zurich, 8091 Zurich, Switzerland; beat.knechtle@hispeed.ch; 12Medbase St. Gallen Am Vadianplatz, 9001 St. Gallen, Switzerland; 13Escola Superior Desporto e Lazer, Instituto Politécnico de Viana do Castelo, Rua Escola Industrial e Comercial de Nun’Álvares, 4900-347 Viana do Castelo, Portugal; filipe.clemente5@gmail.com; 14Instituto de Telecomunicações, Delegação da Covilhã, 1049-001 Lisboa, Portugal; 15Department of Physiology, Federal University of Sergipe (UFS), São Cristóvão 49100-000, Brazil

**Keywords:** spinal cord injury, resistance training, oxidative stress, muscle damage

## Abstract

**Simple Summary:**

Spinal Cord Injury is a devastating condition that compromises the individual’s health, quality of life and functional independence. Rats submitted to Spinal Cord Injury were evaluated after four weeks of resistance training. Analyses of levels of muscle damage and oxidative stress surgery were performed. Resistance training demonstrated increase antioxidative activity while decreased oxidative damage in injured rats, in addition to having presented changes in the levels of muscle damage in that same group. The results highlight that resistance training promoted a decrease in oxidative stress and a significant response in muscle damage markers.

**Abstract:**

Background: Spinal cord injury (SCI) is a condition that affects the central nervous system, is characterized by motor and sensory impairments, and impacts individuals’ lives. The objective of this study was to evaluate the effects of resistance training on oxidative stress and muscle damage in spinal cord injured rats. Methodology: Forty Wistar rats were selected and divided equally into five groups: Healthy Control (CON), Sham (SHAM) SCI Untrained group (SCI-U), SCI Trained group (SCI- T), SCI Active Trained group (SCI- AT). Animals in the trained groups were submitted to an incomplete SCI at T9. Thereafter, they performed a protocol of resistance training for four weeks. Results: Significant differences in muscle damage markers and oxidative stress in the trained groups, mainly in SCI- AT, were found. On the other hand, SCI- U group presented higher levels of oxidative stress and biomarkers of LDH and AST. Conclusion: The results highlight that resistance training promoted a decrease in oxidative stress and a significative response in muscle damage markers.

## 1. Introduction

Spinal cord injury (SCI) is a devastating condition that affects the central nervous system and is characterized by motor and sensory impairments. It also impacts individual quality of life including physical, psychological, economic and social aspects [1,2]. The practice of physical exercise for functional recovery has been widely used and its mechanisms have been studied by researchers around the world, such as Fu et al. [3].

The evidence has demonstrated the exercise benefits in the general physiological and psychosocial functions of SCI population [4,5,6], which include locomotor function, mood, pain, and life satisfaction improvement [3,7,8]. Exercise as a treatment strategy for people with spinal cord injury has shown that residual functions have been preserved or improved in individuals trained in conventional ways or with partial support of body weight [9,10].

A recently published physical activity guideline recommends that individuals with incomplete spinal cord injury perform at least three sets of muscle strengthening exercises since there is a correlation between the muscle strength of the hip flexors, extensors and abductors and better walking performance [11,12]. There is still no definitive treatment for spinal cord injury, but many studies have been carried out, highlighting here pre-clinical studies which allow the investigation of disease-related mechanisms, along with new treatment alternatives for this serious health condition.

Studies using animal models of exercise have been shown to be useful in understanding the central nervous system and the regulation of exercise performance, and biological and biochemical analyses provide a means of investigating these responses. [13,14,15] Understanding the evolution process of the primary lesion as well as the responses to exercise can contribute to the development of more effective therapeutic strategies [16,17].

The effects of resistance training (RT) after SCI were the main objective of this study. We used a rat model of SCI partial transection, enough to produce a deficit but to allow weight-bearing exercise.

## 2. Materials and Methods

### 2.1. Study Design

Weeks 1 and 2: The animals were trained during the first week, without apparatus, for approximately 30 min/day, and in the second week were trained with an empty apparatus in the tail;

Weeks from 3 to 6: Animals in the training groups were trained to climb a vertical staircase with the loads tied to their tails, and rested between the series for 2 min in a dark box at the top of the stairs. Study design is shown in Table 1.

### 2.2. Animal Care and Ethics Committee Approval

Forty adult male Wistar rats weighing 150–180 g were obtained from the Animal Center of the Federal University of Sergipe. The animals were maintained at 21 ± 2 °C with food and water ad libitum under a 12:12 h light/dark cycle. All experimental procedures were approved by the Ethics Committee for Animal Use in Research of the Federal University of Sergipe (protocol number 36/2017) and were conducted in compliance with the Guide for Care and Use of Laboratory Animals (National Institutes of Health) [18].

### 2.3. Experimental Groups

Animals were randomly divided into five groups, each one with eight animals (*n* = 8): Healthy Control (CON), constituted by rats not exposed to any surgical procedure or treatment, Sham (SHAM), surgical control animals submitted to laminectomy surgery without SCI, SCI Untrained group (SCI-U), animals that underwent partial transection but were untrained, SCI Trained group (SCI-T), animals that underwent partial transection and were trained, and SCI Active Trained group (SCI-AT), animals that trained previously, underwent partial transection and were trained again.

### 2.4. Surgical Procedure

The animals were anaesthetized with ketamine and xylazine at a dose of 75 and 14 mg/kg of body weight, respectively, and administrated intraperitonially as verified by the absence of the tail and paw withdrawal of the pinching reflex. After the trichotomy and aseptic measures, the animals were submitted to surgery to remove spinous process and expose spinal cord at the thoracic vertebra 9 (T9). A laminectomy and then a partial transection at the right side using micro scissors and a scalpel was performed and then the muscles and skin were sutured in layers. Postoperative treatment included antibiotic therapy (Pencivet^®^) intramuscularly to small animals [19]. The animals were placed in an appropriate box, isolated from other animals and accommodated in an air-conditioned room for post-surgical recovery. The general health of all animals was verified, and the bladder and intestine manually emptied until the restoration of normal conditions of urination and defection.

### 2.5. Training Protocol

Two weeks before surgery, all rats, except control group (CG) trained on the staircase for adaptation. On the first week, animals trained without apparatus in the tail, for approximately 30 min/day, and in the second week they trained with an empty apparatus in the tail. After the period of adaptation to the device, rats in the training groups were trained to climb a 1.1-m vertical staircase (60° incline) with the loads tied to their tails. At the top of the stairs there was a dark box (20 × 20 × 20 cm) that allowed the animals to rest between the series, and the adopted rest interval was 2 min. Protocol consisted of three sessions a week, 30 min per day (time corresponding to three sets of eight repetitions each) with a total training time period of four weeks.

Before each training, animals of SCI-T and SCI-AT groups were individually weighed to adjust loads. The weight lifted was fixed at 25% of the animal’s weight in the first week and increased to 75% in the last week.

### 2.6. Euthanasia and Collection of Biological Material

After one hour of the last training, the animals were weighted and anesthetized via intraperitoneal injection of ketamine and xylazine (10 and 85 mg/kg, respectively). Already under the effect of the anesthetic, the animals were sacrificed by decapitation with a guillotine. Then, brain, heart, liver, right gastrocnemius muscle and triceps were collected, in addition to blood for further analysis.

After the collection, the blood sample was immediately centrifuged at 4000× *g* for 15 min at 4 °C and the supernatant was stored at −80 °C. The other organs were washed three times with 1.15% potassium chloride (KCl) solution, dried, and weighed. Thus, they were homogenized, wherein each gram of the tissue was mixed with 5 mL KCl + 10 µL phenylmethyl sulfonyl fluoride (100 mmol/L) + 15 µL 10% Triton solution and then centrifuged at 3000× *g* for 10 min at 4 °C and the supernatant stored at −80 °C.

### 2.7. Tissue Damage Analysis

Quantification of muscle damage was evaluated by enzymatic tissue damage markers such as Creatine Kinase (CK), Lactate Dehydrogenase (LDH), Alanine Aminotransferase (ALT), and Aspartate Aminotransferase (AST). A commercial kit (Labtest^®^, Lagoa Santa, Minas Gerais, Brazil) was used, 20 µL of each animal homogenized in specific reagents at 37 ± 0.2 °C, and readings taken using a spectrophotometer (Biospectro Model SP-22 UV/Visible, Curitiba, Brazil) at a wavelength of 340 nm.

### 2.8. Oxidative Stress (OS) Analysis

The lipid oxidation was determined by measuring Thio-barbituric Acid Reactive Substances (TBARS), according to the method described by Lapenna [20]. We used aliquots of 200 µL of the samples (blood and tissues) added to a 400 µL solution of trichloroacetic acid (TCA; 15%), HCl (0.25 N), TBA (0.375%), and butylated hydroxytoluene (BHT; 2.5 mM), 4 µL of sodium dodecyl sulfate (8.1%), heated for 30 min at 95 °C in an oven. The pH of the solution was adjusted to 0.9 with concentrated HCl. To prevent lipid peroxidation during heating, BHT was used. After cooling the solution to room temperature, 4 mL butanol was added, followed by centrifugation at 800× *g* for 15 min at 4 °C. Next, the absorbance of the supernatant was measured at 532 nm. A molar extinction coefficient of 1.54 10^5^/M/cm was used. The TBARS results are expressed malondialdehyde (MDA) equivalents (nmol MDA eq/mL) for plasma and tissue samples.

Following the methodology described by Faure and Lafond [21], the antioxidant activity in the plasma and tissues was quantified through the sulfhydryl (SH) groups. Briefly, aliquots of 50 µL of samples were mixed with 1 mL of tris-EDTA buffer (pH 8.2). The absorbance (A1) was measured at 412 nm. The samples were then transferred to test tubes containing 20 µL DTNB (10 mM), diluted in methanol (4 mg/mL), and left undisturbed in a dark room. After 15 min, the absorbance (A2) was measured. The SH concentration was calculated using the following equation: (A2 − A1) − B × 1.57 mM × 1000, and the result was expressed in nmol/mg tissue.

### 2.9. Statistical Analysis

After confirmation of normality through the test of Shapiro Wilk and homogeneity assumptions, one-way ANOVA with Bonferroni’s post hoc was performed to compare the measurements groups. To check the effect size, partial Eta squared (η^2^p) was used, adopting values of low effect (≤0.05), medium effect (0.05 to 0.25), high effect (0.25 to 0.50), and very high effect (>0.50) for ANOVA [22]. The level of significance was set at *p* < 0.05. Data are presented as means (X) ± standard deviation (SD). All statistical analyses were performed using the computerized package Statistical Package for the Social Science (SPSS), version 22.0 (IBM Corp, Armonk, NY, USA).

## 3. Results

The results of muscle damage (CK, LDH, ALT and AST) are shown in Figure 1.

There was no difference in CK and ALT among the groups. In relation to the LDH there was difference between CON and SCI-U, and in relation to SCI-AT groups (*p* < 0.001, F(4,28) = 88.118; η^2^p = 0.926, very high effect). Regarding AST levels, there were differences between SHAM and SCI- AT groups (*p* = 0.001) and SCI-T (*p* < 0.001; F(4,28) = 5.275; η^2^p = 0.433, high effect).

Sulfhydryl levels (SH) were analyzed in the blood and in the following tissues: brain, heart, liver, gastrocnemius and triceps muscles (Figure 2).

For blood concentration of SH, there were differences between SCI- AT and SCI- U groups (*p* = 0.021) and between SCI- AT and CON groups (*p* = 0.036; F(4,28) = 8.452; η^2^p = 0.547) (very high effect). There were differences in SH brain levels between SCI-T and SCI-U groups (*p* = 0.001) and SCI- T and SHAM (*p* = 0.015, F(4,28) = 7.150; η^2^p = 0.505, very high effect). In heart SH analysis, there were differences between SHAM and SCI- AT groups (*p* = 0.019; F(4,28) = 7.781; η^2^p = 0.526 (high effect). There were differences in SH liver levels between CON and SHAM groups (*p* < 0.001) and CON and SCI-T (*p* = 0.006, F(4,28) = 9.813; η^2^p = 0.584, very high effect). In liver SH analysis, there were differences between SHAM and SCI- AT groups (*p* = 0.019; F(4,28) = 7.781; η^2^p = 0.526 (high effect). There were no differences between the groups in relation to gastrocnemius and triceps muscles.

Malondialdehyde (MDA) levels analysis in the blood and in the following tissues: brain, heart, liver, gastrocnemius and triceps muscles, are shown in Figure 3.

Malondialdehyde (MDA) levels analysis showed differences between CON and SCI-U groups (*p* = 0.031; F(4,28) = 7.194; η^2^p = 0.507) (very high effect). Regarding MDA levels in the brain, there were differences between CON and SCI-U groups (*p* = 0.028) and CON and SHAM groups (*p* = 0.035, F(4,28) = 3.685; η^2^p = 0.343, high effect). In heart MDA levels, there were differences between CON and SCI-U (*p* = 0.005), CON and SHAM (*p* = 0.008) and CON and SCI-T (*p* = 0.045; F(4,28) = 10.511; η^2^p = 0.600 (very high effect). Regarding liver MDA levels, there were differences between CON and SCI-U groups (*p* = 0.048), CON and SHAM (*p* = 0.023) and CON and SCI-T groups (*p* = 0.018; F(4,28) = 9.814; η^2^p = 0.584 (very high effect). In Tríceps MDA levels, there were differences between the groups CON and SHAM (*p* = 0.020) and CON and SCI-T (*p* = 0.033; F(4,28) = 6.973; η^2^p = 0.499 (high effect), while in gastrocnemius there was no difference between groups.

## 4. Discussion

This study aimed to analyze the effect of resistance training on muscle damage and oxidative stress in spinal cord injured rats, by biochemical indicators in the blood, brain, heart, liver, gastrocnemius and triceps tissues. The results highlighted that there was a decrease in oxidative stress and a significant response in muscle damage markers among the groups.

Significant differences in muscle damage and OS were found in the SCI-AT group compared with the SCI-T group. This suggests that four weeks of resistance training induces positive responses following spinal cord injury and ameliorates deleterious effects and so should be considered a potential strategy for SCI treatment.

The link between exercise, muscle damage and oxidative stress in different populations has been widely studied [23,24]. However, the impact of exercise on biomarkers in people with SCI was only recently brought to light. The importance of studying this relation is due to the fact that muscle wasting after SCI is associated with changes in body composition which contributes to increased risk of cardiovascular disease and type 2 diabetes, interfering in individual health and quality of life [25].

It is already known that the release of inflammatory mediators after SCI alters neural function, which impairs ionic conduction and synaptic transmission. Thus, knowing the possible mechanisms with which to control this inflammatory response can contribute to the reduction of damage caused by SCI. One of the ways to carry out this analysis is through biomarkers, which determine the evolution of the lesion and also point out the results of applied therapeutic strategies [26].

In the analysis of tissue damage, there are methods that use analysis of biomarkers such as creatine kinase (CK), lactate dehydrogenase (LDH), alanine aminotransferase (ALT), and aspartate aminotransferase (AST), which together estimate the potential exercise injury. The results revealed no differences in Creatine Kinase (CK) levels among the groups after 4 weeks of training. Different from the findings here, in the study of Dos Santos et al. rats submitted to squat exercise presented a significant increase in CK after four weeks of training and this may be related to the intense activity of healthy muscles of animals. On the other hand, the study of Mohammadi et al. [27] found no difference in levels of CK, in groups of healthy subjects submitted to different volumes of eccentric exercise.

Creatine Kinase is an enzyme located in skeletal muscle cells, considered an indirect, highly sensitive and specific indicator of muscle damage with a direct relationship between this and muscle activity, and is impaired in people with SCI, suggesting that the protocol training did not generate enough muscular contractions to generate damage in trained animals [28,29,30,31].

The results for Alanine Aminotransferase (ALT) and Aspartate Aminotransferase (AST) were different. While ALT did not present differences among the groups, AST presented differences between SHAM and SCI-AT groups. Both are important liver enzymes, ALT found in higher concentration in cytoplasm whose elevation is acute and directly proportional to the injury. The absence of significative difference may suggest that there was no liver damage after the protocol of RT. Concerning AST, this is an enzyme found in other tissues besides liver such as kidney, brain and skeletal muscle, therefore intense exercise to which the animals were submitted may explain the changes in levels. [32,33,34].

Partially supporting our results, studies performed with other exercise protocols with healthy animals showed no significant differences in ALT and AST levels. In our study, while ALT presented no differences among the groups, AST levels presented differences between CON and SCI- AT groups. In the study of Motta et al. [35], a protocol of swimming with weights ALT and AST levels presented a decrease, while another showed no differences in these markers. Low levels of ALT and AST can provide evidence that exercise did not cause hepatic damage [31,36].

In addition to muscle damage analysis, LDH levels presented differences in SCI-AT group compared with CON and SHAM groups. The increase in LDH levels is related to cellular damage, being better used as a damage confirmation marker rather than a damage extension marker [37]. Its levels increase more slowly than CK, but stay increased longer. LDH is a marker of skeletal muscle injuries due to micro-rupture of muscle fibers and shows the degree of metabolic adaptation of skeletal muscles [38]. Considering that the higher levels occurred in a group that was submitted to the resistance training protocol before the injury (SCI- AT) was different from the others, we can suggest that this group had evidence of muscle damage [39].

Since the discovery that prolonged exercise promotes oxidative stress, many studies have been conducted and have shown that exercise increases reactive oxygen species (ROS) which results in oxidative stress and can be measured by the analysis of oxidative damage in various tissues, including blood and skeletal muscle [40]. However, it must be remembered that regular physical exercise increases the regulation of the enzymatic antioxidant system and the modulation of oxidative damage through the regulation of the cellular redox state by modulating the metabolism in an intensity-dependent manner and/or by direct activation of ROS-generating enzymes [41]. Regular exercise can even promote an increase in the brain’s antioxidant capacity [42,43] and the increase in hippocampal neurogenesis [44], among other effects.

Resulting from the SCI, rapid skeletal muscle atrophy occurs in the following six weeks, this period marked by an increase of oxidative stress (OS), while plasma antioxidant levels are decreased. Therefore, OS decrease or inhibition can reduce the deleterious effects of SCI [45].

Concerning the analysis of OS parameters, sulfhydryl (SH) and Malondialdehyde (MDA) levels were used. We observed that RT promoted a significant difference in SH levels in different tissues in the trained groups (SCI- AT and SCI-T) compared with the others (SCI-U, SHAM and CON groups), results that evidence the role of exercise, corroborating studies that demonstrated that exercise training can promote a decrease in oxidative stress, and also increase the antioxidant system [45,46].

Being considered an indirect antioxidant biomarker, SH is found in the GSH Cys residue and other antioxidants, whereby molecules stabilize free radicals by receiving their unpaired electron. The results in this study are divergent from de Araujo [47], in which Wistar rats were submitted to a protocol of high intensity interval training (HIIT) but did not find significant differences in SH levels [48,49].

The results showed that SCI induction caused a significant increase in blood, brain, heart, liver and triceps levels of Malondialdehyde (MDA), mainly SCI-U group. Considered the primary biomarker of oxidative damage, this is the result of many chemical reactions that occur in lipidic peroxidation, in short, the degradation of cell membrane by the action of reactive species, whose measurement is made by the formation of MDA through its complex action with thio-barbituric acid [50,51].

The lower levels of MDA in trained animals suggests a reduction of lipidic peroxidation. On the other hand, if we consider the higher levels of SH in these groups, we can deduce that there is a trend towards an increase of antioxidant activity. It is already known that exercise improves the capacity of the cell antioxidant defense system in order to neutralize ROS increases, in addition to improving the metabolic state, resulting in a redox balance, although the intensity necessary to produce this balance has not yet been found [52,53,54].

An important aspect that should be mentioned is that different responses of muscle damage and oxidative stress markers were found in the different tissues analyzed and this is due to the fact that each organ responds differently to exercise. [55]

Nevertheless, despite these important findings, a limitation of the present study was the absence of an apparatus to accurately control the extent of the lesion, although the transection model is widely used in the literature. The Basso-Beatie-Bresnahan (BBB) functionality assessment scale was used in the postoperative period to ensure sample homogeneity and minimize these differences between groups.

It is important to highlight another limitation, related to the results of the SHAM group, which was exposed to the surgical procedure and laminectomy but did not suffer spinal cord injury. Based on the results found in this study, there was an alteration in the levels of several markers, which may be related to the stress of surgery, as well as the handling of animals. We can also infer that the invasive procedure, even without injury, may have triggered some physiological response that raised these parameters.

Furthermore, concerning spinal cord injury and the surgical procedure to which the animals were submitted, some aspects should be considered. In this sense, the different segments of the spinal cord end up by interfering in specific disease outcomes [56,57]. The spinal cord is a complex and dynamic neural structure, whose neurons, when injured, tend to interfere with the generation of sympathetic activity in many autonomic targets, including the heart and blood vessels [58,59], with evidence of the interaction between the spinal cord. and the heart, [60,61], and it can interfere in several activities, which were not the target of our study. Despite these possible influences that spinal cord injury tends to promote, we consider these points as secondary to our study, and the focus of research is only in terms of exercise-related spinal cord injury.

## 5. Conclusions

To the best of our knowledge, there are no similar studies involving SCI, resistance training, oxidative stress and muscle damage.

Spinal Cord Injury is known to affect all aspects of an individual’s life as a result of decreased function associated with loss of skeletal muscle mass. Resistance training groups demonstrated reduce MDA levels compared with non-trained animals while increasing SH levels in the same groups. Thus, resistance training provides a potential strategy for reducing the deleterious effects of muscle damage and oxidative stress in individuals with SCI.

## Figures and Tables

**Figure 1 biology-11-00032-f001:**
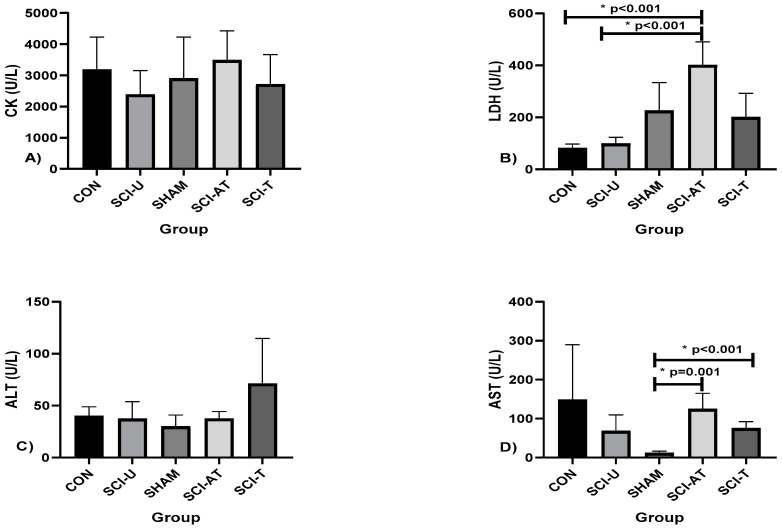
Muscle Damage levels presented by (**A**) Creatine Kinase (CK), (**B**) Lactate Dehydrogenase (LDH), (**C**) Alanine Aminotransferase (ALT), (**D**) Aspartate Aminotransferase (AST) after interventions in the different groups: Control (CON), constituted by rats not exposed to any surgical procedure or treatment, Sham (SHAM), surgical control animals submitted to laminectomy surgery without SCI, SCI Untrained group (SCI-U), animals that underwent partial transection but were untrained, SCI Trained group (SCI- T), animals that underwent partial transection and were trained, SCI Active Trained group (SCI- AT), animals that trained previously, underwent partial transection and were trained again. * *p* < 0.05 ANOVA (one-way), and Bonferroni post hoc.

**Figure 2 biology-11-00032-f002:**
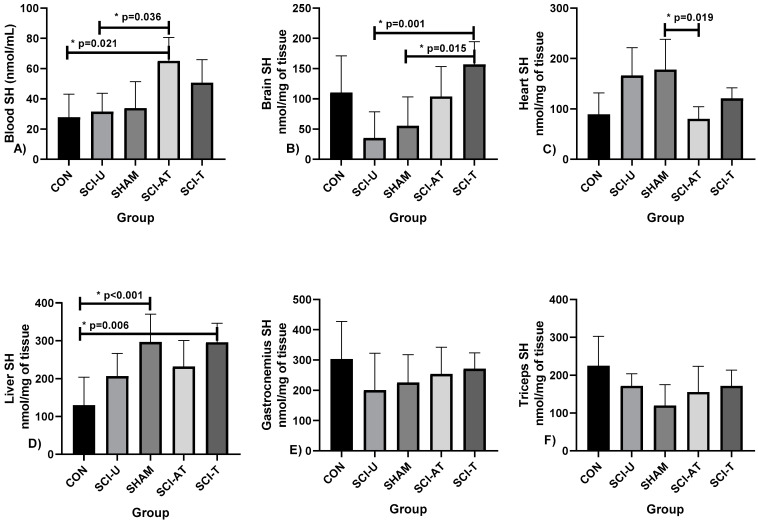
Oxidative Stress measured by Sulfhydryl (SH) levels measured in: (**A**) Blood, (**B**) Brain, (**C**) Heart, (**D**) Liver, (**E**) Gastrocnemius, (**F**) Triceps after interventions in the different groups: Control (CON), constituted by rats not exposed to any surgical procedure or treatment, Sham (SHAM), surgical control animals submitted to laminectomy surgery without SCI, SCI Untrained group (SCI-U), animals that underwent partial transection but were untrained, SCI Trained group (SCI- T), animals that underwent partial transection and were trained, SCI Active Trained group (SCI- AT), animals that trained previously, underwent partial transection and were trained again. * *p* < 0.05 ANOVA (one-way), and Bonferroni post hoc.

**Figure 3 biology-11-00032-f003:**
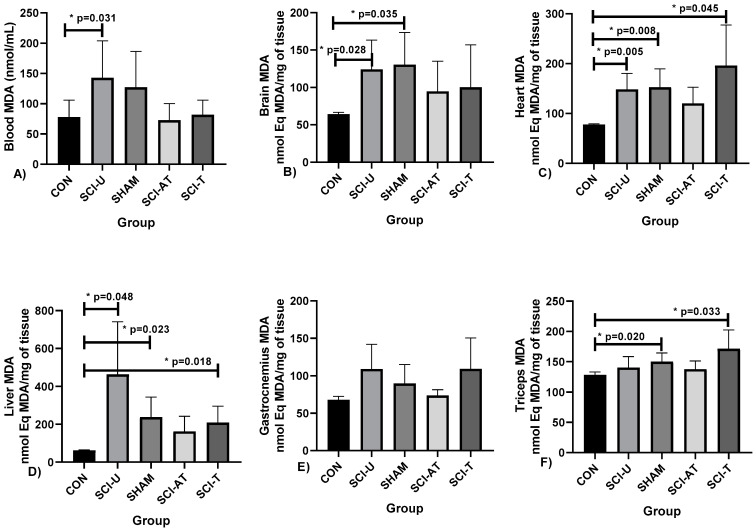
Malondialdehyde (MDA) levels measured in: (**A**) Blood, (**B**) Brain, (**C**) Heart, (**D**) Liver, (**E**) Gastrocnemius, (**F**) Triceps after interventions in the different groups: Control (CON), constituted by rats not exposed to any surgical procedure or treatment, Sham (SHAM), surgical control animals submitted to laminectomy surgery without SCI, SCI Untrained group (SCI-U), animals that underwent partial transection but were untrained, SCI Trained group (SCI- T), animals that underwent partial transection and were trained, SCI Active Trained group (SCI- AT), animals that trained previously, underwent partial transection and were trained again. * *p* < 0.05 ANOVA (one-way), and Bonferroni post hoc.

**Table 1 biology-11-00032-t001:** Experimental drawing. Weekly training schedule.

Familiarization		Training
Week 1 1 × 30 min (3 × Week)	Week 2 1 × 30 min (3 × Week)	Week 3 to 6 (3 × Week) 3 × 30 min [Rest between sets: 2 min]
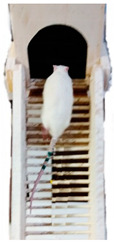	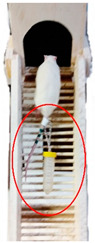	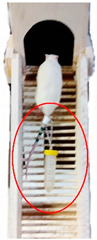

## Data Availability

The data that support this study can be obtained from the address: www.ufs.br/Department of Physical Education. Accessed on 16 October 2021.

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
