# Peer review of "Effects of Resistance Training on Oxidative Stress Markers and Muscle Damage in Spinal Cord Injured Rats"

_biology, 2021, doi:10.3390/biology11010032_

Round 1

Reviewer 1 Report

This study assesses the use of resistance training on oxidative stress markers and muscle damage in spinal cord injured rats.

Line 57: Delete "purpose"

LIne 57-58: This statement suggests many researchers, however only one reference provided. Please add references or amend the sentence.

Line 58: Change "has" to "have"

Line 68-71. This sentence is difficult to follow. Please rephrase.

Line 74: Change "analyzes" to analyses"

Line 74: Change "...biochemical analyses are means to find these responses..." to "...biochemical analyses provide a means of investigating these responses..." This sentence is also too long.

Line 77: Delete "Then" and add "In this study we used..."

Table 1 legend is confusing. Is this Table 1 or Figure 1? The first figure in the Results section is also Figure 1. I think it is best to keep this as Table 1. The legend should be changed to make the description clearer.

Line 82: Change "...trained on the first..." to "...trained during the first..."

Line 202: Change "Regarding to bloody SH,..." to "With regard to blood concentration of SH,..."

Line 210: Change "...are in figure 3." to "...are shown in Figure 3."

Line 233 & 234: Change "significative" to "significant"

Line 234: It is not good to start a sentence with an acronym. Please rephrase this sentence.

Line 234-235: Change to"Significant differences in muscle damage and OS were found in the SCI-AT group compared with the SCI-T group. This suggests that four weeks of resistance training induces positive responses following spinal cord injury and ameliorates deleterious effects and so, it should be considered a potential strategy for SCI treatment."

Line 238-239: Change "population" to "populations"

Line 254: Change "founds" to "findings"

Line 255: Change "significantive" to "significant"

Line 257: Change "...indifferent levels..." to "...no difference in levels..."

Line 258: Change "levels" to "volumes"

Lines 263-264: Change to " The results for Alanine Aminotransferase (ALT) and Aspartate Aminotransferase (AST) were different."

Line 270: Change "Supporting partially..." to "Partially supporting..."

Line 270: Change "...another protocols..." to "...other protocols..."

Line 274: Change "swim" to "swimming"

Line 278: Change "...compared to..." to "...compared with..."

Line 279: Change "...increase slowly than CK..." to "...increase more slowly than CK..."

Line 280: Change "...for longer," to "...for longer. Also, delete"what can explain the results found,"

Line 283: Change "...muscle damage evidence." to "...evidence of muscle damage."

Line 285: Change "carried" to "conducted" and "showed" to "have shown"

Line 288: Change "As result of the SCI. a rapid..." to "Resulting from the SCI, rapid..."

Line 289: Delete "being"

Line 292: Delete "to"

Line 294: Change "compared to" to "compared with"

Lines 295-296: Change "corroborating with studies, that..." to "corroborating studies that..."

Line 300: Reference number for de Araujo is missing.

Line 315: Change "the present study has as limitation..." to " a limitation of the present study..."

Line 317: Delete "e" and put a comma after Beatie.

Line 321: Change "...is no studies..." to "...are no studies..."

Lines 323-324: Change to "Spinal cord injury is known to affect all aspects of an individuals life as a result of decreased function associated with loss of skeletal muscle mass."

Line 325: Change "compared to" to "compared with"

Lines 326-327: Change final sentence to "Thus, resistance training provides a potential strategy for reducing the deleterious effects of muscle damage and oxidative stress in individuals with SCI."

Author Response

Dear Reviewer 1
Initially, thank you for the considerations pointed out. After making all the suggested adjustments, I forward the manuscript. Respectfully.

Reviewer 2 Report

Manuscript 1453540 is interesting and takes into consideration the interplay between oxidative stress, muscle damage and the effect of resistance training in the context of spinal cord injury.
The manuscript also has potential in consideration of the experimental model that was used to evaluate the effect of resistance training.
However, some critical issues do not allow the manuscript to be published in this form.
To validate the authors' conclusions some simple experiments can be conducted and with which the manuscript would certainly be even more complete.

Major revisions.

a) CK is a marker of muscle damage, while LDH is an indicator of cell damage in general (eg blood cells). This needs to be clarified and commented upon in discussion as it changes with training.

b) As far as oxidative stress is concerned, only oxidative damage parameters were evaluated, but enzymatic and non-enzymatic antioxidant systems, which on the other hand are generally induced with training, were not taken into consideration.
I suggest to the authors, since they certainly have a lot of biological sample at their disposal, to test some enzymatic activity (SOD, CAT, etc.) through some commercial kit.

c) There is no explanation for the difference found between ALT and AST. This data must be commented on in relation to the difference found.

d) It is always difficult to comment on the results when there is a great heterogeneity of data between different tissues. This certainly works in the authors' favor. However, the difference in response found in the various tissues must be commented on and justified.

e) In the statistical analysis of some parameters in relation to the different tissues it seems that statistical significance has been lost. For example LDH SHAM vs CTRL, or Brain SH CTRL vs SCI-U and Triceps SH CTRL vs SHAM just to name a few. I suggest the authors reanalyze them all.

f) In general, surgery without SCI (SHAM) seems to have an influence in many parameters analyzed, and although it is a note in favor of the authors, it sometimes makes one lose sight of the effect of the training. For example, in LDH and AST just to name a few. Authors should extend the comment to this limitation.

Minor revisions.

a) Page 2: is it a table 1 or a figure 1?

b) Please check for some typing errors (eg line 300).

c) Please double-check and rephrase the sentence on lines 250-253.

Author Response

Dear Reviewer 2
Initially, thank you for the considerations pointed out. After making all the suggested adjustments, I forward the manuscript. Respectfully.

Round 2

Reviewer 2 Report

The authors responded to most of the issues raised. However, the authors need to review some aspects before the manuscript can be accepted.

a) In the reviewer's response file, sometimes there is a mismatch in the reference of the lines of the review with the text in its latest version. Please check.

b) The authors argue that LDH is a good indicator for assessing muscle damage, lines 294-296. It's not true.

c) Training is certainly a factor that can greatly improve the redox state and antioxidant defenses of animals. This is also important in the particular experimental model used by the authors. However, there is no mention of the adaptive response induced by the training but only the oxidative damage has been analyzed. In discussion it is worth spending a few sentences on this topic.

d) As written by the authors, the surgery alters several parameters. This makes their interpretation difficult in relation to the other experimental groups. Furthermore, the only significant data between SCI-U and SCI-T or SCI-AT relate to -SH blood and brain. A more extensive comment on this aspect must be made in discussion. 

e) There are still some formatting and typing errors (eg SHAW instead of SHAM). 

Author Response

Initially, we would like to thank you for your considerations. We carry out all the suggested adjustments in accordance with the attached documents.

Round 3

Reviewer 2 Report

The authors responded to most of the issues raised. 

In the revised version the authors added a paragraph under discussion (lines 354-361 attention to the point at line 360) and rightly comment that the intervention affects the functions of the spinal cord and this has repercussions on the functionality of various organs.

But how does this clarify that the only significant data relate to SCI-U and SCI-T or SCI-AT in -SH blood and brain?

Please comment on this issue. 

Author Response

Dear Reviewer
I hope everything is alright. We are grateful for the support and considerations pointed out, they greatly helped not only in the improvement of the manuscript, but also in our own personal growth. We try to answer your inquiries and make ourselves available for any other indications or adjustments that you deem pertinent.

Best Regards
